# Advancements in Arabic Grammatical Error Detection and Correction: An Empirical Investigation

**Bashar Alhafni, Go Inoue[†], Christian Khairallah, Nizar Habash**
Computational Approaches to Modeling Language Lab
New York University Abu Dhabi
[†]Mohamed bin Zayed University of Artificial Intelligence
{alhafni,christian.khairallah,nizar.habash}@nyu.edu
go.inoue@mbzuai.ac.ae

## Abstract

Grammatical error correction (GEC) is a well-explored problem in English with many existing models and datasets. However, research on GEC in morphologically rich languages has been limited due to challenges such as data scarcity and language complexity. In this paper, we present the first results on Arabic GEC using two newly developed Transformer-based pretrained sequence-to-sequence models. We also define the task of multi-class Arabic grammatical error detection (GED) and present the first results on multi-class Arabic GED. We show that using GED information as an auxiliary input in GEC models improves GEC performance across three datasets spanning different genres. Moreover, we also investigate the use of contextual morphological preprocessing in aiding GEC systems. Our models achieve SOTA results on two Arabic GEC shared task datasets and establish a strong benchmark on a recently created dataset. We make our code, data, and pretrained models publicly available.[1]

## 1 Introduction

English grammatical error correction (GEC) has witnessed significant progress in recent years due to increased research efforts and the organization of several shared tasks (Ng et al., 2013, 2014; Bryant et al., 2019). Most state-of-the-art (SOTA) GEC systems borrow modeling ideas from neural machine translation (MT) to translate from erroneous to corrected texts. In contrast, grammatical error detection (GED), which focuses on locating and identifying errors in text, is usually treated as a sequence labeling task. Both tasks have evident pedagogical benefits to native (L1) and foreign (L2) language teachers and students. Also, modeling GED information explicitly within GEC systems yields better results in English (Yuan et al., 2021).

When it comes to morphologically rich languages, GEC and GED have not received as much attention, largely due to the lack of datasets and standardized error type annotations. Specifically for Arabic, the focus on GEC started with the QALB-2014 (Mohit et al., 2014) and QALB-2015 (Rozovskaya et al., 2015) shared tasks; however, recent sequence-to-sequence (Seq2Seq) modeling advances have not been explored much in Arabic GEC. Moreover, multi-class Arabic GED has not been investigated due to the lack of error type information in Arabic GEC datasets. In this paper, we try to address these challenges. Our main contributions are as follows:

1. We are the first to benchmark newly developed pretrained Seq2Seq models on Arabic GEC.

2. We tackle the task of Arabic GED by introducing word-level GED labels for existing Arabic GEC datasets, and present the first results on multi-class Arabic GED.

3. We systematically show that using GED information in GEC models improves performance across GEC datasets in different domains.

4. We leverage contextual morphological preprocessing in improving GEC performance.

5. We achieve SOTA results on two (L1 and L2) previously published Arabic GEC datasets. We also establish a strong benchmark on a recently created L1 Arabic GEC dataset.

## 2 Related Work

**GEC Approaches** Early efforts focused on building feature-based machine learning (ML) classifiers to fix common error types (Chodorow et al., 2007; Tetreault and Chodorow, 2008; Dahlmeier and Ng, 2011; Kochmar et al., 2012; Rozovskaya and Roth, 2013; Farra et al., 2014). Such models required feature engineering and lacked the ability to correct all error types simultaneously.

Reformulating GEC as a monolingual MT task alleviated these issues, first with statistical MT ap-

---

[1] https://github.com/CAMeL-Lab/arabic-gec

proaches (Felice et al., 2014; Junczys-Dowmunt and Grundkiewicz, 2014, 2016) and then neural MT approaches (Yuan and Briscoe, 2016; Xie et al., 2016; Junczys-Dowmunt et al., 2018; Watson et al., 2018), with Transformer-based models being the most dominant (Yuan et al., 2019; Zhao et al., 2019; Grundkiewicz et al., 2019; Katsumata and Komachi, 2020; Yuan and Bryant, 2021).

More recently, edit-based models have been proposed to solve GEC (Awasthi et al., 2019; Malmi et al., 2019; Stahlberg and Kumar, 2020; Mallinson et al., 2020; Omelianchuk et al., 2020; Straka et al., 2021; Mallinson et al., 2022; Mesham et al., 2023). While Seq2Seq models generate corrections to erroneous input, edit-based models generate a sequence of corrective edit operations. Edit-based models add explainability to GEC and improve inference time efficiency. However, they generally require human engineering to define the size and scope of the edit operations (Bryant et al., 2023).

**GED Approaches** Rei and Yannakoudakis (2016) presented the first GED results using a neural approach framing GED as a binary (correct/incorrect) sequence tagging problem. Others used pretrained language models (PLMs) such as BERT (Devlin et al., 2019), ELECTRA (Clark et al., 2020), and XLNet (Yang et al., 2019) to improve binary GED (Bell et al., 2019; Kaneko and Komachi, 2019; Yuan et al., 2021; Rothe et al., 2021). Zhao et al. (2019) and Yuan et al. (2019) demonstrated that combining GED and GEC yields improved results: they used multi-task learning to add token-level and sentence-level GED as auxiliary tasks when training for GEC. Similarly, Yuan et al. (2021) showed that binary and multi-class GED improves GEC.

**Arabic GEC and GED** The Qatar Arabic Language Bank (QALB) project (Zaghouani et al., 2014, 2015) organized the first Arabic GEC shared tasks: QALB-2014 (L1) (Mohit et al., 2014) and QALB-2015 (L1 and L2) (Rozovskaya et al., 2015). Recently, Habash and Palfreyman (2022) created the ZAEBUC corpus, a new L1 Arabic GEC corpus of essays written by university students. We report on all of these sets.

Arabic GEC modeling efforts ranged from feature-based ML classifiers to statistical MT models (Rozovskaya et al., 2014; Bougares and Bouamor, 2015; Nawar, 2015). Watson et al. (2018) introduced the first character-level Seq2Seq

| Dataset | Split | Words | Err. | Type | Domain |
|---------|-------|-------|------|------|--------|
| **QALB-2014** | Train-L1 | 1M | 30% | L1 | Comments |
| | Dev-L1 | 54K | 31% | L1 | Comments |
| | Test-L1 | 51K | 32% | L1 | Comments |
| **QALB-2015** | Train-L2 | 43K | 30% | L2 | Essays |
| | Dev-L2 | 25K | 29% | L2 | Essays |
| | Test-L2 | 23K | 29% | L2 | Essays |
| | Test-L1 | 49K | 27% | L1 | Comments |
| **ZAEBUC** | Train-L1 | 25K | 24% | L1 | Essays |
| | Dev-L1 | 5K | 25% | L1 | Essays |
| | Test-L1 | 5K | 26% | L1 | Essays |

Table 1: Corpus statistics of Arabic GEC datasets.

model and achieved SOTA results on the L1 Arabic GEC data used in the QALB-2014 and 2015 shared tasks. Recently, vanilla Transformers (Vaswani et al., 2017) were explored for synthetic data generation to improve L1 Arabic GEC and were tested on the L1 data of the QALB-2014 and 2015 shared tasks (Solyman et al., 2021, 2022, 2023). To the best of our knowledge, the last QALB-2015 L2 reported results were presented in the shared task itself. We compare our systems against the best previously developed models whenever feasible.

A number of researchers reported on Arabic binary GED. Habash and Roth (2011) used feature-engineered SVM classifiers to detect Arabic handwriting recognition errors. Alkhatib et al. (2020) and Madi and Al-Khalifa (2020) used LSTM-based classifiers. None of them used any of the publicly available GEC datasets mentioned above to train and test their systems. In our work, we explore multi-class GED by obtaining error type annotations from ARETA (Belkebir and Habash, 2021), an automatic error type annotation tool for MSA. To our knowledge, we are the first to report on Arabic multi-class GED. We report on publicly available data to enable future comparisons.

## 3 Background

### 3.1 Arabic Linguistic Facts

Modern Standard Arabic (MSA) is the official form of Arabic primarily used in education and media across the Arab world. MSA coexists in a **diglossic** (Ferguson, 1959) relationship with local Arabic dialects that are used for daily interactions. When native speakers write in MSA, there is frequent code-mixing with the dialects in terms of phonological, morphological, and lexical choices (Habash et al., 2008). In this paper, we focus on MSA GEC. While its **orthography** is standardized, written Arabic suffers many orthographic inconsistencies even

in professionally written news articles (Buckwalter, 2004; Habash et al., 2012). For example, hamzated Alifs (أ Â, إ Ǎ)[2] are commonly confused with the un-hamzated letter (ا A), and the word-final letters ي y and ى ý are often used interchangeably. These errors affect 11% of all words (4.5 errors per sentence) in the Penn Arabic Treebank (Habash, 2010). Additionally, the use of punctuation in Arabic is very inconsistent, and omitting punctuation marks is very frequent (Awad, 2013; Zaghouani and Awad, 2016). Punctuation errors constitute ∼40% of errors in the QALB-2014 GEC shared task. This is ten times higher than punctuation errors found in the English data used in the CoNLL-2013 GEC shared task (Ng et al., 2013). Arabic has a large vocabulary size resulting from its rich **morphology**, which inflects for gender, number, person, case, state, mood, voice, and aspect, and cliticizes numerous particles and pronouns. Arabic's diglossia, orthographic inconsistencies, and morphological richness pose major challenges to GEC models.

### 3.2 Arabic GEC Data

We report on three publicly available Arabic GEC datasets. The first two come from the **QALB-2014** (Mohit et al., 2014) and **QALB-2015** (Rozovskaya et al., 2015) shared tasks. The third is the newly created **ZAEBUC** dataset (Habash and Palfreyman, 2022). None of them were manually annotated for specific error types. Table 1 presents a summary of the dataset statistics. Detailed dataset statistics are presented in Appendix B.

QALB-2014 consists of native/L1 user comments from the Aljazeera news website, whereas QALB-2015 consists of essays written by Arabic L2 learners with various levels of proficiency. Both datasets have publicly available training (Train), development (Dev), and test (Test) splits. The ZAEBUC dataset comprises essays written by native Arabic speakers, which were manually corrected and annotated for writing proficiency using the Common European Framework of Reference (CEFR) (Council of Europe, 2001). Since the ZAEBUC dataset did not have standard splits, we randomly split it into Train (70%), Dev (15%), and Test (15%), while keeping a balanced distribution of CEFR levels.

The three sets vary in a number of dimensions: domain, level, number of words, percentage of erroneous words, and types of errors. Appendix C

---

presents automatic error type distributions over the training portions of the three datasets. Orthographic errors are more common in the L1 datasets (QALB-2014 and ZAEBUC) compared to the L2 dataset (QALB-2015). In contrast, morphological, syntactic, and semantic errors are more common in QALB-2015. Punctuation errors are more common in QALB-2014 and QALB-2015 compared with ZAEBUC.

### 3.3 Metrics for GEC and GED

GEC systems are most commonly evaluated using reference-based metrics such as the MaxMatch ($M^2$) scorer (Dahlmeier and Ng, 2012), ERRANT (Bryant et al., 2017), and GLEU (Napoles et al., 2015), among other reference-based and reference-less metrics (Felice and Briscoe, 2015; Napoles et al., 2016; Asano et al., 2017; Choshen et al., 2020; Maeda et al., 2022). In this work, we use the $M^2$ scorer because it is language agnostic and was the main evaluation metric used in previous work on Arabic GEC. The $M^2$ scorer compares hypothesis edits made by a GEC system against annotated reference edits and calculates the precision (P), recall (R), and $F_{0.5}$. In terms of GED, we follow previous work (Bell et al., 2019; Kaneko and Komachi, 2019; Yuan et al., 2021) and use macro precision (P), recall (R), and $F_{0.5}$ for evaluation. We also report accuracy.

## 4 Arabic Grammatical Error Detection

Most of the work on GED has focused on English (§2), where error type annotations are provided manually (Yannakoudakis et al., 2011; Dahlmeier et al., 2013) or obtained automatically using an error type annotation tool such as ERRANT (Bryant et al., 2017). However, when it comes to morphologically rich languages such as Arabic, GED remains a challenge. This is largely due to the lack of manually annotated data and standardized error type frameworks. In this work, we treat GED as a mutli-class sequence labeling task. We present a method to automatically obtain error type annotations by extracting edits from parallel erroneous and corrected sentences and then passing them to an Arabic error type annotation tool. To the best of our knowledge, this is the first work that explores multi-class GED in Arabic.

### 4.1 Edit Extraction

Before automatically labeling each erroneous sentence token, we need to align the erroneous and

---

[2]Arabic HSB transliteration (Habash et al., 2007).

| Alignments | | 13 | 12 | 11 | 10 | 9 | 8 | | 7 | 6 | 5 | 4 | 3 | 2 | 1 | |
|---|---|---|---|---|---|---|---|---|---|---|---|---|---|---|---|---|
| | **Erroneous** | وإيجابيه | سلبية | منها | اثار | لها | ف | | بحكمه | الإجتماعي | التواصل | وسائل | إستخدام | من | لابد | |
| | | wAyjAbyh | slbyh | mnhA | AθAr | lhA | f | | bHkmh | AlÄjtmAςy | AltwASl | wsAŷl | ÄstxdAm | mn | lAbd | |
| | | 14 | 13 | 12 | 11 | 10 | 9 | 8 | 7 | 6 | 5 | 4 | 3 | 2 | 1 | |
| | **Corrected** | . | وإيجابية | سلبية | آثار | فلها | ، | بحكمة | الاجتماعي | التواصل | وسائل | استخدام | من | بد | لا | |
| | | | wAyjAbyh | slbyh | ÄθAr | flhA | | bHkmh | AlAjtmAςy | AltwASl | wsAŷl | AstxdAm | mn | bd | lA | |

| Edits | | | | | | | | | | | | | | | |
|---|---|---|---|---|---|---|---|---|---|---|---|---|---|---|---|
| | **M²** | | | | R | | | | | | K | K | R | K | S |
| | **Lev.** | R | | M | | M | | R | R | | K | K | R | K | S |
| | **ARETA** | R | K | D | R | R | R | R | R | | K | K | R | K | S |
| | **Ours** | I | R | K | D | M | I | R | R | | K | K | R | K | S |

| Error Type | | | | | | | | | | | | | | | |
|---|---|---|---|---|---|---|---|---|---|---|---|---|---|---|---|
| | **43-Class** | PM | OH+OT | Delete | OH | Merge | PM | OT | OH | | | | OH | | Split |
| | **13-Class** | P | O | Delete | O | Merge | P | O | O | | | | O | | Split |
| | **2-Class** | E | E | E | E | E | E | E | E | | | | E | | E |

Figure 1: An example showing the differences between the alignments of the M² scorer, a standard Levenshtein distance, ARETA, and our proposed algorithm. The edit operations are keep (**K**), replace (**R**), insert (**I**), delete (**D**), merge (**M**), and split (**S**). Dotted lines between the erroneous and corrected sentences represent gold alignment. The last three rows present different granularities of ARETA error types based on our alignment. The sentence in the figure can be translated as "*Social media must be used wisely, as it has both negative and positive effects*".

corrected sentence pairs to locate the positions of all edits so as to map errors to corrections. This step is usually referred to as *edit extraction* in GEC literature (Bryant et al., 2017).

We first obtain character-level alignment between the erroneous and corrected sentence pair by computing the weighted Levenshtein edit distance (Levenshtein, 1966) for each pair of tokens in the two sentences. The output of this alignment is a sequence of token-level edit operations representing the minimum number of insertions, deletions, and replacements needed to transform one token into another. Each of these operations involves one token at most belonging to either sentence. However, some errors may involve more than one single edit operation. To capture multi-token edits, we extend the alignment to cover merges and splits by implementing an iterative algorithm that greedily merges or splits adjacent tokens such that the overall cumulative edit distance is minimized.

### 4.2 Error Type Annotation

Next, we pass the extracted edits to an automatic annotation tool to label them with specific error types. We use ARETA, an automatic error type annotation tool for MSA (Belkebir and Habash, 2021). Internally, ARETA is built using a combination of rule-based components and an Arabic morphological analyzer (Taji et al., 2018; Obeid et al., 2020). It uses the error taxonomy of the Arabic Learner Corpus (ALC) (Alfaifi and Atwell, 2012; Alfaifi

| | QALB-2014 | | | QALB-2015 | | |
|---|---|---|---|---|---|---|
| | P↑ | R↑ | AER↓ | P↑ | R↑ | AER↓ |
| **M²** | 92.5 | 87.1 | 0.10 | 90.8 | 83.3 | 0.13 |
| **Lev.** | 86.8 | 84.3 | 0.14 | 84.5 | 84.2 | 0.16 |
| **ARETA** | 84.3 | 82.9 | 0.16 | 84.1 | 84.7 | 0.16 |
| **Ours** | **99.6** | **99.7** | **0.00** | **97.7** | **98.0** | **0.02** |

Table 2: Evaluation of different alignment algorithms.

et al., 2013) which defines seven error classes covering orthography (**O**), morphology (**M**), syntax (**X**), semantics (**S**), punctuation (**P**), merges, and splits. The error classes are further differentiated into 32 error tags that can be assigned individually or in combination.

ARETA comes with its own alignment algorithm that extracts edits, however, it does not handle many-to-one and many-to-many edit operations (Belkebir and Habash, 2021). We replace ARETA's internal alignment algorithm with ours to increase the coverage of error typing. Using our edit extraction algorithm with ARETA enables us to automatically annotate single-token and multi-token edits with various error types. Appendix C presents the error types obtained from ARETA by using our alignment over the three GEC datasets we use.

To demonstrate the effectiveness of our alignment algorithm, we compare our algorithm to the alignments generated by the M² scorer, a standard Levenshtein edit distance, and ARETA. Table 2 presents the evaluation results of the alignment algorithms against the manual gold alignments of the

QALB-2014 and QALB-2015 Dev sets in terms of precision (P), recall (R), and alignment error rate (AER) (Mihalcea and Pedersen, 2003; Och and Ney, 2003). Results show that our alignment algorithm is superior across all metrics.

Figure 1 presents an example of the different alignments generated by the algorithms we evaluated. The $M^2$ scorer's alignment over-clusters multiple edits into a single edit (words 6–13). This is not ideal, particularly because the $M^2$ scorer does not count partial matches during the evaluation, which leads to underestimating the models' performances (Felice and Briscoe, 2015). A standard Levenshtein alignment does not handle merges correctly, e.g., words 8 and 9 in the erroneous sentence are aligned to words 9 and 10 in the corrected version. Among the drawbacks of ARETA's alignment is that it does not handle merges, e.g., erroneous words 8 and 9 are aligned with corrected words 9 and 10, respectively.

## 5 Arabic Grammatical Error Correction

Recently developed GEC models rely on Transformer-based architectures, from standard Seq2Seq models to edit-based systems built on top of Transformer encoders. Given Arabic's morphological richness and the relatively small size of available data, we explore different GEC models, from morphological analyzers and rule-based systems to pretrained Seq2Seq models. Primarily, we are interested in exploring modeling approaches to address the following two questions:

- **RQ1**: Does morphological preprocessing improve GEC in Arabic?

- **RQ2**: Does modeling GED explicitly improve GEC in Arabic?

**Morphological Disambiguation (Morph)** We use the current SOTA MSA morphological analyzer and disambiguator from CAMeL Tools (Inoue et al., 2022; Obeid et al., 2020). Given an input sentence, the analyzer generates a set of potential analyses for each word and the disambiguator selects the optimal analysis in context. The analyses include minimal spelling corrections for common errors, diacritizations, POS tags, and lemmas. We use the dediacritized spellings as the corrections.

**Maximum Likelihood Estimation (MLE)** We exploit our alignment algorithm to build a simple lookup model to map erroneous words to their corrections. We implement this model as a bigram

maximum likelihood estimator over the training data: $P(c_i|w_i, w_{i-1}, e_i)$; where $w_i$ and $w_{i-1}$ are the erroneous word (or phrases in case of a merge error) and its bigram context, $e_i$ is the error type of $w_i$, and $c_i$ is the correction of $w_i$. During inference, we pick the correction that maximizes the MLE probability. If the bigram context ($w_i$ and $w_{i-1}$) was not observed during training, we backoff to a unigram. If the erroneous input word was not observed in training, we pass it to the output.

**ChatGPT** Given the rising interest in using large language models (LLMs) for a variety of NLP tasks, we benchmark ChatGPT (GPT-3.5) on the task of Arabic GEC. We follow the setup presented by Fang et al. (2023) on English GEC. To the best of our knowledge, we are the first to present ChatGPT results on Arabic GEC. The experimental setup along with the used prompts are presented in Appendix A.

**Seq2Seq with GED Models** We experiment with two newly developed pretrained Arabic Transformer-based Seq2Seq models: **AraBART** (Kamal Eddine et al., 2022) (pretrained on 24GB of MSA data mostly in the news domain), and **AraT5** (Nagoudi et al., 2022) (pretrained on 256GB of both MSA and Twitter data).

We extend the Seq2Seq models we use to incorporate token-level GED information during training and inference. Specifically, we feed predicted GED tags as auxiliary input to the Seq2Seq models. We add an embedding layer to the encoders of AraBART and AraT5 right after their corresponding token embedding layers, allowing us to learn representations for the auxiliary GED input. The GED embeddings have the same dimensions as the positional and token embeddings, so all three embeddings can be summed before they are passed to the multi-head attention layers in the encoders.

Our approach is similar to what was done by Yuan et al. (2021), but it is much simpler as it reduces the model's size and complexity by not introducing an additional encoder to process GED input. Since the training data we use is relatively small, not drastically increasing the size of AraBART and AraT5 becomes important not to hinder training.

## 6 Experiments

### 6.1 Arabic Grammatical Error Detection

We build word-level GED classifiers using Transformer-based PLMs. From the many avail-

| | | 43-Class | | | | 13-Class | | | | 2-Class | | | |
|---|---|---|---|---|---|---|---|---|---|---|---|---|---|
| | | **P** | **R** | **$F_{0.5}$** | **Acc.** | **P** | **R** | **$F_{0.5}$** | **Acc.** | **P** | **R** | **$F_{0.5}$** | **Acc.** |
| **QALB-2014** | Dev-L1 | 56.7 | 48.4 | 53.3 | 94.1 | 69.0 | 58.7 | 65.3 | 94.7 | 95.8 | 92.7 | 95.1 | 96.1 |
| | Test-L1 | 55.0 | 45.5 | 50.6 | 93.6 | 58.1 | 54.2 | 56.8 | 94.1 | 95.4 | 91.5 | 94.5 | 95.5 |
| **QALB-2015** | Dev-L2 | 39.0 | 35.0 | 36.9 | 84.5 | 55.1 | 47.3 | 51.7 | 85.3 | 87.0 | 80.4 | 85.2 | 88.9 |
| | Test-L1 | 51.8 | 45.3 | 49.4 | 94.9 | 66.5 | 56.2 | 60.7 | 95.6 | 96.2 | 93.9 | 95.7 | 96.7 |
| | Test-L2 | 37.0 | 35.4 | 35.8 | 85.5 | 52.8 | 48.6 | 51.0 | 86.5 | 88.6 | 81.3 | 86.6 | 89.9 |
| **ZAEBUC** | Dev-L1 | 50.9 | 43.7 | 47.5 | 92.6 | 57.1 | 52.9 | 55.7 | 93.3 | 95.7 | 92.8 | 95.1 | 95.5 |
| | Test-L1 | 54.9 | 43.3 | 49.8 | 91.9 | 69.2 | 56.6 | 62.4 | 92.6 | 95.5 | 92.5 | 94.8 | 95.2 |

Table 3: GED results on the Dev and Test sets in terms of macro precision, recall, $F_{0.5}$, and accuracy.

able Arabic monolingual BERT models (Antoun et al., 2020; Abdul-Mageed et al., 2021; Lan et al., 2020; Safaya et al., 2020; Abdelali et al., 2021), we chose to use CAMeLBERT MSA (Inoue et al., 2021), as it was pretrained on the largest MSA dataset to date.

In our GED modeling experiments, we project multi-token error type annotations to single-token labels. In the case of a Merge error (many-to-one), we label the first token as *Merge-B* (Merge beginning) and all subsequent tokens as *Merge-I* (Merge inside). For all other multi-token error types, we repeat the same label for each token. We further label all deletion errors with a single *Delete* tag. To reduce the output space of the error tags, we only model the 14 most frequent error combinations (appearing more than 100 times). We ignore unknown errors when we compute the loss during training; however, we penalize the models for missing them in the evaluation. Since the majority of insertion errors are related to missing punctuation marks rather than missing words (see Appendix C), and due to inconsistent punctuation error annotations (Mohit et al., 2014), we exclude insertion errors from our GED modeling and evaluation. We leave the investigation of insertion errors to future work. The full GED output space we model consists of 43 error tags (43-Class).

We take advantage of the modularity of the ARETA error tags to conduct multi-class GED experiments, reducing the 43 error tags to their corresponding 13 main error categories as well as to a binary space (correct/incorrect). The statistics of the error tags we model across all datasets are in Appendix D. Figure 1 shows an example of error types at different granularity levels. Table 3 presents the GED granularity results. Unsurprisingly, all numbers go up when we model fewer error types. However, modeling more error types does not significantly worsen the performance in terms of error detection accuracy. It seems that all systems are capable of detecting comparable

numbers of errors despite the number of classes, but the verbose systems struggle with detecting the specific class labels.

## 6.2 Arabic Grammatical Error Correction

We explore different variants of the above-mentioned Seq2Seq models. For each model, we study the effects of applying morphological preprocessing (**+Morph**), providing GED tags as auxiliary input (**+GED**), or both (**+Morph+GED**). Applying morphological preprocessing simply means correcting the erroneous input using the morphological disambiguator before training and inference.

To increase the robustness of the models that take GED tags as auxiliary input, we use predicted (not gold) GED tags when we train the GEC systems. For each dataset, we run its respective GED model on the same training data it was trained on and we pick the predictions of the *worst* checkpoint. During inference, we resolve merge and delete errors before feeding erroneous sentences to the model. This experimental setup yields the best performance across all GEC models.

To ensure fair comparison to previous work on Arabic GEC, we follow the same constraints that were introduced in the QALB-2014 and QALB-2015 shared tasks: systems tested on QALB-2014 are only allowed to use the QALB-2014 training data, whereas systems tested on QALB-2015 are allowed to use the QALB-2014 and QALB-2015 training data. For ZAEBUC, we train our systems on the combinations of the three training datasets. We report our results in terms of precision (P), recall (R), $F_1$, and $F_{0.5}$. $F_1$ was the official metric used in the QALB-2014 and QALB-2015 shared tasks. However, we follow the most recent work on GEC and use $F_{0.5}$ (weighing precision twice as much as recall) as our main evaluation metric.

We use Hugging Face's Transformers (Wolf et al., 2019) to build our GED and GEC models. The hyperparameters we used are detailed in Appendix A.

| | QALB-2014 | | | | QALB-2015 | | | | ZAEBUC | | | | Avg. |
|---|---|---|---|---|---|---|---|---|---|---|---|---|---|
| | **P** | **R** | **F$_1$** | **F$_{0.5}$** | **P** | **R** | **F$_1$** | **F$_{0.5}$** | **P** | **R** | **F$_1$** | **F$_{0.5}$** | **F$_{0.5}$** |
| **B&B (2015)** | - | - | - | - | 56.7 | 34.8 | 43.1 | 50.4 | - | - | - | - | - |
| **W+ (2018)** | 80.0 | 62.5 | 70.2 | 75.8 | - | - | - | - | - | - | - | - | - |
| **Morph** | 76.4 | 30.4 | 43.5 | 58.7 | 56.2 | 9.4 | 16.2 | 28.2 | 78.0 | 36.9 | 50.1 | 63.8 | 50.2 |
| **MLE** | **89.2** | 41.3 | 56.5 | 72.4 | **73.7** | 20.1 | 31.6 | 48.0 | **90.1** | 55.6 | 68.8 | 80.1 | 66.9 |
|   +Morph | 88.5 | 44.9 | 59.6 | 74.1 | 68.3 | 22.0 | 33.2 | 48.0 | 89.1 | 61.8 | 73.0 | 81.9 | 68.0 |
| **ChatGPT** | 67.7 | 60.6 | 63.9 | 66.1 | 54.9 | 36.9 | 44.1 | 50.0 | 68.1 | 52.1 | 59.1 | 64.2 | 60.1 |
| **AraT5** | 82.5 | 66.3 | 73.5 | 78.6 | 69.3 | 39.4 | 50.2 | 60.2 | 84.1 | 67.4 | 74.8 | 80.1 | 73.0 |
|   +Morph | 83.1 | 65.8 | 73.4 | 78.9 | 69.7 | 40.6 | 51.3 | 60.9 | 85.0 | 71.3 | 77.5 | 81.8 | 73.9 |
|   +GED[43] | 82.6 | 67.1 | 74.1 | 79.0 | 69.5 | 41.9 | 52.3 | 61.4 | 85.7 | 66.7 | 75.0 | 81.0 | 73.8 |
|   +Morph +GED[43] | 83.1 | **67.9** | **74.7** | **79.6** | 68.4 | 41.5 | 51.7 | 60.6 | 85.2 | 71.2 | 77.6 | 82.0 | 74.0 |
| **AraBART** | 83.2 | 64.9 | 72.9 | 78.7 | 68.6 | 42.6 | 52.6 | 61.2 | 87.3 | 70.6 | 78.1 | 83.4 | 74.4 |
|   +Morph | 82.4 | 67.2 | 74.0 | 78.8 | 68.5 | 44.3 | 53.8 | 61.7 | 87.2 | 71.6 | 78.7 | 83.6 | 74.7 |
|   +GED[43] | 83.3 | 65.9 | 73.6 | 79.1 | 68.2 | 45.3 | 54.4 | 61.9 | 87.2 | 72.9 | 79.4 | 83.9 | 75.0 |
|   +Morph +GED[43] | 83.4 | 66.3 | 73.9 | 79.3 | 68.2 | **46.6** | **55.4** | **62.4** | 87.3 | **73.6** | **79.9** | **84.2** | **75.3** |

Table 4: GEC results on the Dev sets of QALB-2014, QALB-2015, and ZAEBUC. B&B (2015) and W+ (2018) refer to Bougares and Bouamor (2015) and Watson et al. (2018), respectively. The best results are in bold.

| | | QALB-2014 | | | | QALB-2015 | | | | ZAEBUC | | | | Avg. |
|---|---|---|---|---|---|---|---|---|---|---|---|---|---|---|
| | | **P** | **R** | **F$_1$** | **F$_{0.5}$** | **P** | **R** | **F$_1$** | **F$_{0.5}$** | **P** | **R** | **F$_1$** | **F$_{0.5}$** | **F$_{0.5}$** |
| 43-Class | [Oracle] | **85.5** | **73.3** | **79.0** | **82.8** | **73.9** | **57.2** | **64.5** | **69.8** | **89.8** | 82.0 | **85.7** | **88.1** | **80.2** |
| 13-Class | [Oracle] | 85.4 | 73.2 | 78.8 | 82.6 | 73.5 | 55.9 | 63.5 | 69.2 | 89.4 | **82.2** | **85.7** | 87.9 | 79.9 |
| 2-Class | [Oracle] | 84.2 | 72.1 | 77.7 | 81.4 | 71.6 | 54.5 | 61.9 | 67.4 | 86.6 | 80.0 | 83.2 | 85.2 | 78.0 |
| 43-Class | | 83.4 | 66.3 | 73.9 | 79.3 | 68.2 | **46.6** | **55.4** | **62.4** | 87.3 | 73.6 | 79.9 | 84.2 | 75.3 |
| 13-Class | | **83.9** | 65.7 | 73.7 | **79.5** | 68.0 | **46.6** | 55.3 | 62.3 | **87.6** | 73.9 | **80.2** | **84.5** | **75.4** |
| 2-Class | | 82.5 | 67.3 | 74.2 | 79.0 | **68.3** | 45.0 | 54.3 | 61.9 | 86.0 | 72.3 | 78.6 | 82.9 | 74.6 |

Table 5: GED granularity results when used within the best GEC system (AraBART+Morph+GED) on the Dev sets of QALB-2014, QALB-2015, and ZAEBUC. The best results are in bold.

## 7 Results

Table 4 presents the results on the Dev sets.

**Baselines** The Morph system which did not use any training data constitutes a solid baseline for mostly addressing the noise in Arabic spelling. The MLE system claims the highest precision of all compared systems, but it suffers from low recall as expected. ChatGPT has the highest recall among the baselines, but with lower precision. A sample of 100 ChatGPT mismatches reveals that 37% are due to mostly acceptable punctuation choices and 25% are valid paraphrases or re-orderings; however, 38% are grammatically or lexically incorrect.

**Seq2Seq Models** AraT5 and AraBART outperform previous work on QALB-2014 and QALB-2015, with AraBART being the better model on average.

**Does morphological preprocessing improve Arabic GEC?** Across all models (MLE, AraT5, and AraBART), training and testing on morphologically preprocessed text improves the performance, except for MLE+Morph on QALB-2015 where there is no change in F$_{0.5}$.

**Does GED help Arabic GEC?** We start off by using the most fine-grained GED model (43-Class) to exploit the full effect of the ARETA GED tags and to guide our choice between AraBART and AraT5. Using GED as an auxiliary input in both AraT5 and AraBART improves the results across all three Dev sets, with AraBART+GED demonstrating superior performance compared to the other models, on average. Applying morphological preprocessing as well as using GED as an auxiliary input yields the best performance across the three Dev sets, except for QALB-2015 in the case of AraT5+Morph+GED. Overall, **AraBART+Morph+GED** is the best performer on average in terms of F$_{0.5}$. The improvements using GED with GEC systems are mostly due to recall. An error comparison between AraBART and the AraBART+Morph+GED model (Appendix E) shows improved performance on the majority of the error types.

To study the effect of GED granularity on GEC, we train two additional AraBART+Morph+GED models with 13-Class and 2-Class GED tags. The results in Table 5 show that 13-Class GED was best in QALB-2014 and ZAEBUC, whereas 43-Class

| | QALB-2014 | | | | QALB-2015-L1 | | | | QALB-2015-L2 | | | | ZAEBUC | | | | Avg. |
| | P | R | F$_1$ | F$_{0.5}$ | P | R | F$_1$ | F$_{0.5}$ | P | R | F$_1$ | F$_{0.5}$ | P | R | F$_1$ | F$_{0.5}$ | F$_{0.5}$ |
|---|---|---|---|---|---|---|---|---|---|---|---|---|---|---|---|---|---|
| **B&B (2015)** | - | - | - | - | - | - | - | - | 54.1 | 33.3 | 41.2 | 48.1 | - | - | - | - | - |
| **W+ (2018)** | - | - | 70.4 | - | - | - | 73.2 | - | - | - | - | - | - | - | - | - | - |
| **S+ (2022)** | 79.1 | 65.8 | 71.8 | 76.0 | 78.4 | 70.4 | 74.2 | 76.6 | - | - | - | - | - | - | - | - | - |
| **AraBART** | 84.0 | 64.7 | 73.1 | 79.3 | 82.0 | 71.7 | 76.5 | 79.7 | **69.6** | 43.5 | 53.5 | 62.1 | **86.0** | 71.6 | 78.2 | 82.7 | 75.9 |
| +Morph | 83.3 | **67.4** | **74.5** | 79.5 | 81.7 | 73.0 | 77.1 | 79.8 | 68.7 | 43.6 | 53.3 | 61.6 | 85.3 | 71.8 | 78.0 | 82.3 | 75.8 |
| +GED[43] | **84.2** | 65.4 | 73.6 | **79.6** | 81.2 | 72.4 | 76.5 | 79.3 | 69.0 | **45.4** | **54.7** | 62.5 | 85.4 | 72.6 | 78.5 | 82.5 | 76.0 |
| +Morph+GED[43] | 83.9 | 65.7 | 73.7 | 79.5 | **82.6** | 72.1 | 77.0 | **80.3** | 67.6 | 45.2 | 54.2 | 61.5 | 85.4 | **73.7** | 79.1 | 82.7 | 76.0 |
| +GED[13] | 84.1 | 65.0 | 73.3 | 79.4 | 81.5 | 72.7 | 76.8 | 79.5 | 69.3 | 44.9 | 54.5 | **62.5** | 85.9 | 73.4 | **79.2** | **83.1** | **76.1** |
| +Morph+GED[13] | 83.9 | 65.3 | 73.4 | 79.4 | 81.1 | 73.4 | 77.1 | 79.5 | 68.2 | 44.8 | 54.1 | 61.8 | 85.2 | **73.7** | 79.0 | 82.6 | 75.8 |
| +GED[2] | 83.8 | 64.5 | 72.9 | 79.1 | 81.4 | 71.5 | 76.2 | 79.2 | 69.1 | 44.9 | 54.4 | 62.4 | 85.7 | 71.5 | 78.0 | 82.4 | 75.8 |
| +Morph+GED[2] | 83.0 | 67.0 | 74.1 | 79.2 | 81.3 | **73.8** | **77.4** | 79.7 | 68.1 | 45.3 | 54.4 | 61.9 | 85.7 | 72.4 | 78.5 | 82.7 | 75.9 |

Table 6: GED granularity results when used within GEC on the Test sets of QALB-2014, QALB-2015, and ZAEBUC. B&B (2015), W+ (2018), and S+ (2022) refer to Bougares and Bouamor (2015), Watson et al. (2018), and Solyman et al. (2022), respectively. The best results are in bold.

| | QALB-2014 | | | | QALB-2015-L1 | | | | QALB-2015-L2 | | | | ZAEBUC | | | | Avg. |
| | P | R | F$_1$ | F$_{0.5}$ | P | R | F$_1$ | F$_{0.5}$ | P | R | F$_1$ | F$_{0.5}$ | P | R | F$_1$ | F$_{0.5}$ | F$_{0.5}$ |
|---|---|---|---|---|---|---|---|---|---|---|---|---|---|---|---|---|---|
| **AraBART** | 89.5 | 77.3 | 83.0 | 86.8 | **90.1** | 81.4 | 85.5 | 88.2 | **71.8** | 40.7 | 52.0 | 62.3 | 89.5 | 76.9 | 82.7 | 86.6 | 81.0 |
| +Morph | 88.4 | 78.9 | 83.4 | 86.3 | 89.9 | 83.1 | 86.4 | 88.5 | 70.2 | 41.8 | 52.4 | 61.8 | 88.4 | 76.3 | 81.9 | 85.7 | 80.6 |
| +GED[43] | 89.7 | 78.9 | 84.0 | **87.3** | 89.8 | 81.8 | 85.6 | 88.1 | 70.7 | **43.6** | **53.9** | **62.9** | 89.2 | 77.0 | 82.7 | 86.5 | 81.2 |
| +Morph+GED[43] | 88.8 | **80.1** | **84.2** | 86.9 | 90.0 | 83.8 | **86.8** | **88.7** | 69.0 | **43.6** | 53.4 | 61.8 | 88.7 | **78.4** | 83.2 | 86.4 | 80.9 |
| +GED[13] | **89.8** | 78.9 | 84.0 | **87.3** | 89.8 | 82.2 | 85.8 | 88.2 | 71.0 | 42.8 | 53.4 | 62.7 | **89.9** | 77.8 | **83.4** | **87.2** | **81.4** |
| +Morph+GED[13] | 88.6 | 80.0 | 84.1 | 86.7 | 89.5 | **84.1** | 86.7 | 88.3 | 68.9 | 43.5 | 53.3 | 61.7 | 88.9 | **78.4** | 83.3 | 86.5 | 80.8 |
| +GED[2] | 89.3 | 77.6 | 83.0 | 86.7 | 89.4 | 81.8 | 85.5 | 87.8 | 70.6 | 42.4 | 53.0 | 62.3 | 89.0 | 77.0 | 82.6 | 86.3 | 80.8 |
| +Morph+GED[2] | 87.8 | 79.8 | 83.6 | 86.1 | 89.9 | 83.0 | 86.3 | 88.5 | 69.5 | 43.5 | 53.5 | 62.1 | 89.2 | 77.7 | 83.1 | 86.6 | 80.8 |

Table 7: No punctuation GED granularity results when used within GEC on the Test sets of QALB-2014, QALB-2015, and ZAEBUC. The best results are in bold.

GED was best in QALB-2015 in terms of F$_{0.5}$. However, in terms of precision and recall, GED models with different granularity behave differently across the three Dev sets. On average, using any GED granularity improves over AraBART, with 13-Class GED yielding the best results, although it is only 0.1 higher than 43-Class GED in terms of F$_{0.5}$. For completeness, we further estimate an oracle upper bound by using gold GED tags with different granularity. The results (in Table 5) show that using GED with different granularity improves the results considerably. This indicates that GED is providing the GEC system with additional information; however, the main bottleneck is the GED prediction reliability as opposed to GED granularity. Improving GED predictions will most likely lead to better GEC results.

**Test Results** Since the best-performing models on the three Dev sets benefit from different GED granularity when used with AraBART+Morph, we present the results on the Test sets using all different GED granularity models. The results of using AraBART and its variants on the Test sets are presented in Table 6. On QALB-2014, using Morph, GED, or both improves the results over AraBART, except for 2-Class GED. AraBART+43-Class GED is the best performer (0.3 increase in F$_{0.5}$, although not statistically significant).[3] It is worth noting that AraBART+Morph achieves the highest recall on QALB-2014 (2.7 increase over AraBART and statistically significant at $p < 0.05$). For QALB-2015-L1, using GED by itself across all granularity did not improve over AraBART, but when combined with Morph, the 43-Class GED model yields the best performance in F$_{0.5}$ (0.6 increase statistically significant at $p < 0.05$). When it comes to QALB-2015-L2, Morph does not help, but using GED alone improves the results over AraBART, with 43-Class and 13-Class GED being the best (0.4 increase). Lastly, in ZAEBUC, Morph does not help, but using 13-Class GED by itself improves over AraBART (0.4 increase). Overall, all the improvements we observe are attributed to recall, which is consistent with the Dev results.

Following the QALB-2015 shared task (Rozovskaya et al., 2015) reporting of no-punctuation

---

[3]Statistical significance was done using a two-sided approximate randomization test.

results due to observed inconsistencies in the references (Mohit et al., 2014), we present results on the Test sets without punctuation errors in Table 7. The results are consistent with those with punctuation, indicating that GED and morphological preprocessing yield improvements compared to using AraBART by itself across all Test sets. The score increase among all reported metrics when removing punctuation, specifically in the L1 data, indicates that punctuation presents a challenge for GEC models and needs further investigation both in terms of data creation and modeling approaches.

**Analyzing the Test Results** Table 8 presents the average absolute changes in precision and recall over the Test sets when introducing Morph, GED, or both. Adding Morph alone or GED alone improves recall (up to 0.8 in the case of Morph) and slightly hurts precision. When using both Morph and GED, we observe significant improvements in recall with an average of 1.5 but with higher drops of precision with an average of $-0.7$.

# 8 Conclusion and Future Work

We presented the first results on Arabic GEC using Transformer-based pretrained Seq2Seq models. We also presented the first results on multi-class Arabic GED. We showed that using GED information as an auxiliary input in GEC models improves GEC performance across three datasets. Further, we investigated the use of contextual morphological preprocessing in aiding GEC systems. Our models achieve SOTA results on two Arabic GEC shared tasks datasets and establish a strong benchmark on a recently created dataset.

In future work, we plan to explore other GED and GEC modeling approaches, including the use of syntactic models (Li et al., 2022; Zhang et al., 2022). We plan to work more on insertions, punctuation, and infrequent error combinations. We also plan to work on GEC for Arabic dialects, i.e., the conventional orthography of dialectal Arabic normalization (Habash et al., 2018; Eskander et al., 2013; Eryani et al., 2020).

# Limitations

Although using GED information as an auxiliary input improves GEC performance, our GED systems are limited as they can only predict error types for up to 512 subwords since they are built by fine-tuning CAMeLBERT. We also acknowledge

|  | P | R |
| --- | --- | --- |
| +Morph | $-0.4$ | 0.8 |
| +GED$^{43}$ | $-0.2$ | 0.7 |
| +GED$^{13}$ | $-0.2$ | 0.7 |
| +GED$^{2}$ | $-0.3$ | 0.5 |
| +GED$^{*}$ | $\mathbf{-0.2}$ | 0.6 |
| +Morph+GED$^{43}$ | $-0.5$ | 1.3 |
| +Morph+GED$^{13}$ | $-0.8$ | 1.4 |
| +Morph+GED$^{2}$ | $-0.8$ | 1.8 |
| +Morph+GED$^{*}$ | $-0.7$ | **1.5** |

Table 8: Average absolute changes in precision (P) and recall (R) when introducing Morph, GED, or both to AraBART and its variants on the Test sets. GED$^{*}$ indicates the average absolute changes of all models using GED. Bolding highlights the best performance across Morph, GED* and Morph+GED*.

the limitation of excluding insertion errors when modeling GED. Furthermore, our GEC systems could benefit from employing a copying mechanism (Zhao et al., 2019; Yuan et al., 2019), particularly because of the limited training data available in Arabic GEC. Moreover, the dataset sizes of QALB-2015-L2 and ZAEBUC are too small to allow us to test for statistical significance.

# Acknowledgements

We thank Ted Briscoe for helpful discussions and constructive feedback. We acknowledge the support of the High Performance Computing Center at New York University Abu Dhabi. Finally, we wish to thank the anonymous reviewers at EMNLP 2023 for their feedback.

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

## A   Detailed Experimental Setup

**Grammatical Error Detection**   Our GED models were fine-tuned for 10 epochs using a learning rate of 5e-5, a batch size of 32, and a seed of 42. At the end of the fine-tuning, we pick the best checkpoint based on the performance on the Dev sets.

**Grammatical Error Correction**   When using AraBART, we fine-tune the models for 10 epochs by using a learning rate of 5e-5, a batch size of 32, a maximum sequence length of 1024, and a seed of 42. For AraT5, we fine-tune the models for 30 epochs by using a learning rate of 1e-4 and the rest of the hyperparameters are the same as the ones used in AraBART. During inference, we use beam search with a beam width of 5 for all models. At the end of the fine-tuning, we pick the best checkpoint based on the performance on the Dev sets by using the $M^2$ scorer. The $M^2$ scorer suffers from extreme running times in cases where the generated outputs differ significantly from the input. To mitigate this bottleneck, we extend the $M^2$ scorer by introducing a time limit for each sentence during evaluation. If the evaluation of a single generated sentence surpasses this limit, we pass the input sentence to the output without modifications. We use this extended version of the $M^2$ scorer when reporting our results on the Dev sets. When reporting our results on the Test sets, we use the $M^2$ scorer release that is provided by the QALB shared task. We make our extended version of the $M^2$ scorer publicly available.

**ChatGPT**   We start with prompting ChatGPT with a 3-shot prompt. Our exact prompt is the following:

"Please identify and correct any spelling and grammar mistakes in the following sentence indicated by <input> INPUT </input> tag. You need to comprehend the sentence as a whole before gradually identifying and correcting any errors while keeping the original sentence structure unchanged as much as possible.

Afterward, output the corrected version directly without any explanations. Here are some in-context examples:

(1), <input> SRC-1 </input>: <output> TGT-1 </output>.

(2), <input> SRC-2 </input>: <output> TGT-2 </output>.

(3), <input> SRC-3 </input>: <output> TGT-3 </output>.

Please feel free to refer to these examples. Remember to format your corrected output results with the tag <output> Your Corrected Version </output>. Please start: <input> INPUT </input>"

# B   Datasets Statistics

| Dataset | Split | Lines | Words | Err. % | Level | Domain |
|---|---|---|---|---|---|---|
| | Train-L1 | 19,411 | 1,021,165 | 30% | Native | Comments |
| QALB-2014 | Dev-L1 | 1,017 | 53,737 | 31% | Native | Comments |
| | Test-L1 | 968 | 51,285 | 32% | Native | Comments |
| | Train-L2 | 310 | 43,353 | 30% | L2 | Essays |
| QALB-2015 | Dev-L2 | 154 | 24,742 | 29% | L2 | Essays |
| | Test-L2 | 158 | 22,808 | 29% | L2 | Essays |
| | Test-L1 | 920 | 48,547 | 27% | Native | Comments |
| | Train-L1 | 150 | 25,127 | 24% | Native | Essays |
| ZAEBUC | Dev-L1 | 33 | 5,276 | 25% | Native | Essays |
| | Test-L1 | 31 | 5,118 | 26% | Native | Essays |

Table 9: Corpus statistics of Arabic GEC datasets.

## C   Error Types Statistics

| | Tag | Error Description | Example | QALB-2014 | | QALB-2015 | | ZAEBUC | |
|---|---|---|---|---|---|---|---|---|---|
| **Orthography (O)** | OA | Alif, Ya & Alif-Maqsura | علي ← على | 7,627 | 3% | 290 | 2% | 27 | 0% |
| | OC | Char Order | تبرينا ← تربينا | 466 | 0% | 45 | 0% | 30 | 0% |
| | OD | Additional Char | يعدوم ← يدوم | 4,086 | 1% | 283 | 2% | 103 | 2% |
| | OG | Lengthening short vowels | نقيمو ← نقيم | 0 | 0% | 0 | 0% | 0 | 0% |
| | OH | Hamza errors | اكثر ← أكثر | 90,579 | 30% | 1,076 | 8% | 1,905 | 32% |
| | OM | Missing char(s) | سالين ← سائلين | 4,062 | 1% | 361 | 3% | 123 | 2% |
| | ON | Nun & Tanwin Confusion | ثوبن ← ثوبٌ | 0 | 0% | 0 | 0% | 0 | 0% |
| | OR | Char Replacement | مصلنا ← وصلنا | 8,350 | 3% | 762 | 6% | 162 | 3% |
| | OS | Shortening long vowels | أوقت ← أوقات | 0 | 0% | 0 | 0% | 0 | 0% |
| | OT | Ha/Ta/Ta-Marbuta Confusion | مشاركه ← مشاركة | 14,688 | 5% | 54 | 0% | 408 | 7% |
| | OW | Confusion in Alif Fariqa | وكانو ← وكانوا | 1,885 | 1% | 32 | 0% | 12 | 0% |
| | OO | *Other orthographic errors* | - | 1,632 | 1% | 38 | 0% | 148 | 2% |
| **Morphology (M)** | MI | Word inflection | معروف ← عارف | 1,360 | 0% | 400 | 3% | 127 | 2% |
| | MT | Verb tense | تفرحني ← أفرحتني | 76 | 0% | 136 | 1% | 4 | 0% |
| | MO | *Other morphological errors* | - | 15 | 0% | 7 | 0% | 3 | 0% |
| **Syntax (X)** | XC | Case | رائع ← رائعاً | 5,980 | 2% | 279 | 2% | 201 | 3% |
| | XF | Definiteness | السن ← سن | 852 | 0% | 835 | 6% | 51 | 1% |
| | XG | Gender | الغربي ← الغربية | 809 | 0% | 317 | 2% | 86 | 1% |
| | XM | Missing word | على ← Null | 1,375 | 0% | 763 | 6% | 68 | 1% |
| | XN | Number | فكرتي ← أفكاري | 1,107 | 0% | 210 | 2% | 30 | 0% |
| | XT | Unnecessary word | Null←على | 1,047 | 0% | 418 | 3% | 116 | 2% |
| | XO | *Other syntactic errors* | - | 3,270 | 1% | 122 | 1% | 57 | 1% |
| **Semantics (S)** | SF | Conjunction error | سبحان ← فسبحان | 96 | 0% | 46 | 0% | 4 | 0% |
| | SW | Word selection error | من ← عن | 4,711 | 2% | 865 | 7% | 120 | 2% |
| | SO | *Other semantic errors* | - | 380 | 0% | 114 | 1% | 27 | 0% |
| **Punctuation (P)** | PC | Punctuation confusion | قال. ← قال: | 11,361 | 4% | 854 | 7% | 237 | 4% |
| | PM | Missing punctuation | العظيم ← العظيم، | 97,271 | 32% | 2,915 | 22% | 479 | 8% |
| | PT | Unnecessary punctuation | العام, ← العام | 5,553 | 2% | 213 | 2% | 204 | 3% |
| | PO | *Other errors in punctuation* | - | 0 | 0% | 0 | 0% | 0 | 0% |
| **Merge** | MG | Words are merged | لايلزم ← لا يلزم | 15,063 | 5% | 377 | 3% | 849 | 14% |
| **Split** | SP | Words are split | و قال ← وقال | 7,828 | 3% | 80 | 1% | 49 | 1% |
| **Unknown** | UNK | Unkown Errors | الظالمون ← الذين ظلموا | 2,053 | 1% | 303 | 2% | 93 | 2% |
| **Comb.** | - | Error Combinations | انسانيه ← إنسانية | 11,304 | 4% | 848 | 7% | 314 | 5% |
| | | | | 304,886 | | 13,043 | | 6,037 | |

Table 10: The statistics of the error types in the Train sets of QALB-2014, QALB-2015, and ZAEBUC. The error types are based on the extended ALC (Alfaifi et al., 2013) taxonomy as used by Belkebir and Habash (2021).

## D  GED Granularity Data Statistics

| | | | QALB-2014 | | | QALB-2015 | | | | ZAEBUC | | |
|---|---|---|---|---|---|---|---|---|---|---|---|---|
| 2-Class | 13-Class | 43-Class | Train | Dev | Test | Train | Dev | Test-L1 | Test-L2 | Train | Dev | Test |
| E | Delete | Delete | 6,442 | 346 | 540 | 584 | 339 | 250 | 309 | 305 | 64 | 66 |
| | Merge-B | Merge-B | 15,063 | 797 | 795 | 377 | 231 | 625 | 199 | 849 | 180 | 133 |
| | Merge-I | Merge-I | 15,296 | 812 | 807 | 390 | 241 | 629 | 200 | 851 | 180 | 133 |
| | M | M | 30 | 0 | 0 | 14 | 3 | 4 | 4 | 6 | 2 | 2 |
| | | MI | 1,360 | 69 | 59 | 400 | 220 | 56 | 169 | 127 | 30 | 25 |
| | | MT | 76 | 0 | 4 | 136 | 72 | 2 | 40 | 4 | 0 | 1 |
| | M+O | MI+OH | 243 | 17 | 15 | 9 | 9 | 8 | 5 | 7 | 1 | 8 |
| | O | O | 3,255 | 166 | 164 | 75 | 52 | 144 | 70 | 296 | 64 | 68 |
| | | OA | 7,627 | 313 | 252 | 290 | 136 | 514 | 138 | 27 | 4 | 6 |
| | | OC | 466 | 27 | 19 | 45 | 23 | 17 | 26 | 30 | 7 | 7 |
| | | OD | 4,086 | 207 | 204 | 283 | 167 | 146 | 166 | 103 | 24 | 21 |
| | | OH | 90,579 | 4,785 | 4,632 | 1,076 | 599 | 4,499 | 587 | 1,905 | 401 | 451 |
| | | OM | 4,062 | 228 | 217 | 361 | 215 | 188 | 184 | 123 | 23 | 30 |
| | | OR | 8,358 | 425 | 446 | 763 | 415 | 369 | 362 | 162 | 32 | 36 |
| | | OT | 14,688 | 758 | 623 | 54 | 37 | 733 | 26 | 408 | 101 | 138 |
| | | OW | 1,885 | 149 | 107 | 32 | 12 | 77 | 9 | 12 | 4 | 2 |
| | | OA+OH | 480 | 19 | 12 | 4 | 1 | 23 | 0 | 1 | 1 | 1 |
| | | OA+OR | 215 | 8 | 6 | 4 | 4 | 11 | 3 | 0 | 1 | 0 |
| | | OD+OG | 573 | 32 | 32 | 22 | 15 | 23 | 9 | 11 | 4 | 2 |
| | | OD+OH | 317 | 11 | 17 | 13 | 2 | 10 | 2 | 8 | 1 | 1 |
| | | OD+OM | 104 | 4 | 5 | 12 | 7 | 1 | 6 | 0 | 2 | 1 |
| | | OD+OR | 675 | 33 | 26 | 61 | 32 | 22 | 32 | 8 | 2 | 2 |
| | | OH+OM | 2,339 | 134 | 123 | 231 | 106 | 114 | 109 | 54 | 15 | 13 |
| | | OH+OT | 1,468 | 56 | 65 | 2 | 1 | 71 | 1 | 31 | 9 | 9 |
| | | OM+OR | 382 | 15 | 19 | 62 | 27 | 23 | 15 | 17 | 0 | 4 |
| | | OR+OT | 193 | 10 | 7 | 4 | 4 | 2 | 1 | 7 | 0 | 0 |
| | O+X | OH+XC | 323 | 24 | 18 | 6 | 3 | 15 | 2 | 20 | 0 | 4 |
| | P | P | 11,379 | 598 | 687 | 855 | 453 | 446 | 483 | 237 | 51 | 36 |
| | S | S | 536 | 41 | 19 | 188 | 125 | 26 | 103 | 44 | 14 | 21 |
| | | SF | 96 | 5 | 4 | 46 | 33 | 2 | 21 | 4 | 0 | 2 |
| | | SW | 4,804 | 201 | 229 | 887 | 502 | 186 | 422 | 121 | 22 | 28 |
| | X | X | 3,668 | 216 | 182 | 144 | 59 | 161 | 57 | 106 | 26 | 17 |
| | | XC | 5,980 | 373 | 369 | 279 | 180 | 289 | 141 | 201 | 31 | 46 |
| | | XC+XG | 296 | 23 | 40 | 0 | 3 | 1 | 0 | 1 | 0 | 0 |
| | | XC+XN | 500 | 18 | 41 | 29 | 13 | 23 | 9 | 24 | 3 | 3 |
| | | XF | 852 | 63 | 25 | 835 | 494 | 35 | 463 | 51 | 12 | 14 |
| | | XG | 809 | 38 | 30 | 317 | 175 | 35 | 158 | 86 | 20 | 24 |
| | | XM | 225 | 15 | 6 | 151 | 91 | 12 | 68 | 14 | 6 | 3 |
| | | XN | 1,107 | 47 | 41 | 210 | 115 | 47 | 84 | 30 | 9 | 2 |
| | | XT | 155 | 16 | 9 | 46 | 26 | 6 | 24 | 15 | 3 | 4 |
| | Split | Split | 7,828 | 432 | 399 | 80 | 42 | 382 | 34 | 49 | 10 | 10 |
| | UNK | UNK | 6,835 | 331 | 300 | 969 | 454 | 257 | 416 | 361 | 78 | 61 |
| C | C | C | 795,510 | 41,875 | 39,690 | 33,007 | 19,004 | 38,063 | 17,651 | 18,411 | 3,839 | 3,683 |
| | | | 1,021,165 | 53,737 | 51,285 | 43,353 | 24,742 | 48,547 | 22,808 | 25,127 | 5,276 | 5,118 |

Table 11: The statistics of the different GED granularity error types we model across the three datasets. The description of the labels in the 13-Class and 43-Class categories are in Appendix C. For the 2-Class labels, **E** refers to erroneous words and **C** refers to correct words.

# E  Error Analysis on Error Types

| | QALB-2014 | | QALB-2015 | | ZAEBUC | |
|---|---|---|---|---|---|---|
| | AraBART | Best System | AraBART | Best System | AraBART | Best System |
| Delete | 40.1 | **40.8** | 40.5 | **45.3** | 47.5 | **51.9** |
| Merge-B | 91.2 | **93.0** | 82.4 | **86.6** | **96.7** | **96.7** |
| Merge-I | 91.0 | **93.0** | 81.7 | **86.4** | **96.7** | **96.7** |
| M | 24.8 | **27.6** | 37.0 | **40.8** | **48.9** | 48.6 |
| M+O | **54.8** | 37.7 | **17.2** | 15.2 | **100.0** | 55.6 |
| O | 94.0 | **94.3** | **80.3** | 80.2 | 94.0 | **94.4** |
| O+X | 67.7 | **73.9** | 0.0 | 0.0 | 0.0 | 0.0 |
| P | 76.4 | **77.4** | **64.5** | 63.7 | **66.8** | 62.8 |
| S | 43.3 | **44.5** | 33.8 | **34.1** | 36.1 | **40.4** |
| X | 58.5 | **61.1** | 59.6 | **63.9** | 69.5 | **72.9** |
| Split | **87.6** | 87.1 | 78.0 | **78.9** | **88.2** | **88.2** |
| UNK | 50.2 | **57.2** | **37.9** | 35.2 | 57.1 | **63.1** |
| C | 96.2 | **96.8** | 89.9 | **91.4** | 95.4 | **96.1** |
| Macro Avg | 67.4 | **68.0** | 54.1 | **55.5** | **69.0** | 66.7 |

Table 12: Specific error type performance of AraBART and our best system (AraBART+Morph+GED[13]) on average on the Dev sets of QALB-2014, QALB-2015, and ZAEBUC. Results are reported in terms of $F_{0.5}$. The best results are in bold.