# OpenReview forum: "Advancements in Arabic Grammatical Error Detection and Correction: An Empirical Investigation"
_EMNLP/2023/Conference — EMNLP 2023 Main_

### Official Review · Reviewer_XEzF · 2023-08-04

**Soundness:** 4

**Excitement:**

4: Strong: This paper deepens the understanding of some phenomenon or lowers the barriers to an existing research direction.

**Paper Topic And Main Contributions:**

This paper builds a new benchmark for the Arabic grammatical error correction (GEC) area. To build the benchmark, it starts with the grammatical error detection (GED) task. In this section, the paper proposes a new alignment algorithm for error type annotation and achieves better results compared with previous methods. In the GEC section, it finds that the pre-trained model BART can be enhanced with GED and Morphological Disambiguation output, and this combination performs best among most of the settings. It also shows how GED granularity affects final GEC performance.

**Reasons To Accept:**

The paper is well-written and easy to understand; the readers who do not speak Arabic, such as me, can also understand the nature and background of Arabic GEC quickly by reading Sections 2 and 3. The newly proposed alignment algorithm and the strong baseline are essential contributions to the Arabic GEC community.

**Reasons To Reject:**

I only have one concern about the missing explanations of results. In Section 7, Line 509, the paper only describes the results. In Section 5, we can see that AraT5 has a larger pre-trained data size than AraBART but performs worse in most of the situations on the GEC dataset. It would be better to provide some explanations about this kind of result.

**Reproducibility:**

5: Could easily reproduce the results.

**Reviewer Confidence:**

4: Quite sure. I tried to check the important points carefully. It's unlikely, though conceivable, that I missed something that should affect my ratings.

---

> ### Author Rebuttal · Authors · 2023-08-26
>
> Thank you very much for your comments and for pointing out the strengths of our work. Previous work on self-supervised learning in Arabic showed that pretraining BERT-based models on more data does not necessarily lead to better results ([Inoue et al., 2021](https://aclanthology.org/2021.wanlp-1.10.pdf)). [Inoue et al., 2021](https://aclanthology.org/2021.wanlp-1.10.pdf) showed that Arabic variants (i.e., MSA vs. dialects) proximity of pretraining data to fine-tuning task-specific data is more important than pretraining data size. Although AraT5 and AraBART are Seq2Seq models, we hypothesize that a similar conclusion can be observed from our results. Particularly because AraT5 was pretrained on a mix of MSA and dialectical text, whereas AraBART was pretrained on MSA-only text. Given that the GEC datasets we use are in MSA, this explains why AraBART is performing better than AraT5 in most cases, even though it was prertrained on less data.
>
> We hope that our response clarifies your concerns. We will make sure to mention this explanation in the extra page we are allowed to add in the camera-ready version if our paper is accepted.

---

### Official Review · Reviewer_2Jvn · 2023-08-05

**Soundness:** 4

**Excitement:**

4: Strong: This paper deepens the understanding of some phenomenon or lowers the barriers to an existing research direction.

**Paper Topic And Main Contributions:**

This paper focuses on investigating the effectiveness of Grammatical Error Detection (GED) on Arabic Grammatical Error Correction (GEC). For the investigation process, they depend on three datasets (QALB-2014, QALB-2015, and ZAEBUC). They claim that they were the first to benchmark newly developed pretrained Seq2Seq models on Arabic Grammatical Error Correction. As for the GED task, they used their own alignment algorithm which shows that they were superior across all metrics. They could show that using GED information in GEC models improves the performance across GEC datasets. In addition, they discussed the importance of leveraging contextual morphological preprocessing in improving the GEC performance. Their experiments achieved SOTA results on the two (L1 and L2) datasets of (QALB-2014 and QALB-21015).

**Reasons To Accept:**

Grammatical error correction (GEC) is still problematic for morphologically rich languages, such as Arabic, due to the complexity and nature of such languages. Having a paper that focuses on GEC for Arabic, helps the NLP community in developing GEC systems for Arabic.


**Reasons To Reject:**

No reasons

**Reproducibility:**

3: Could reproduce the results with some difficulty. The settings of parameters are underspecified or subjectively determined; the training/evaluation data are not widely available.

**Reviewer Confidence:**

5: Positive that my evaluation is correct. I read the paper very carefully and I am very familiar with related work.

**Typos Grammar Style And Presentation Improvements:**

It would be better if you add the structure of the paper in the last part of the introduction.

---

> ### Author Rebuttal · Authors · 2023-08-26
>
> Thank you very much for your comments and for pointing out the strengths of our work. We will definitely add a paragraph at the end of the introduction outlining the structure of the paper in the camera-ready version if our paper is accepted.

---

### Official Review · Reviewer_51Ai · 2023-08-05

**Soundness:** 3

**Excitement:**

3: Ambivalent: It has merits (e.g., it reports state-of-the-art results, the idea is nice), but there are key weaknesses (e.g., it describes incremental work), and it can significantly benefit from another round of revision. However, I won't object to accepting it if my co-reviewers champion it.

**Paper Topic And Main Contributions:**

The paper addresses the problem of grammatical error correction (GEC) and grammatical error detection (GED) in Arabic, more specifically on Modern Standard Arabic (no dialectal Arabic). Arabic is a morphologically rich language therefore the tasks is more difficult compared to e.g. English.

The authors work with three existing datasets for GEC in MSA, for all of them the erroneous text is provided with the corrected one.

For GED, the authors extend the three existing datasets with automatic annotation of errors types. This mainly requires alignment of the two texts. The authors propose a novel method for obtaining automatic word alignment of erroneous and correct text in Arabic, achieving almost 100% Precision and Recall. After alignment, the errors are automatically classified into 7 error classes and 32 subtypes (assigned individually or in a combination). An Arabic PLM CAMeLBERT is finetuned for the classification tasks with varying number of classes (43/13/2) and evaluated on the three data sets.

For GEC, where each error must be detected and a replacement produced.  The authors proposes two models, one based on AraBART, one on AraT5, each in three configurations, one includes morphological analysis of each Arabic word, one detecting error type, the last one combining both. The performance of those models are compared with other solutions and a few baselines (including e.g. ChatGPT). The results show a superior performance of the models combining both GED and Morphological analysis.

**Questions For The Authors:**

The problem of Arabic dialects is not really discussed in the paper. The large pretrained models are probably trained also on dialectal data. Are dialectal words considered as errors in MSA texts? Is there a special error type for them?

Please, provide better description of the alignment algorithm. It is not clear, whether the Levenshtein algorithm is applied to tokens or to the whole sentences.  The greedy part of the algorithm is not explained well.

Line 565 refers to a statistical test of significance. Please, provide details on the test results also for other experiments (mainly presented in Table 6).



**Reasons To Accept:**

Nicely written paper on an interesting topic. Showing how explicit GED and Morphological analysis can help in GEC in Arabic.
Strong evaluation on three diverse datasets, outperforming the current SOTA. Comparison with several baselines, also including ChatGPT.
The list of references is extremely rich and can well serve as a source of relevant references on this topic


**Reasons To Reject:**

While the authors outperform the current SOTA on all the datasets, the main reason for that is the large pretrained models - AraT5 and AraBART (the latter showed better perfomrance). The effect of including the morhological analysis and error classification is rather small (measured on the test sets, Table 6), most of the results are reported as not statistically significant.

Only the overall (average) improvements of the models combining GAD and Morphological analysis is provided. It is not clear, whether there are some error types where the models can benefit from knowing the explicit information of the type of the error.

It is not clear, from the paper, how acurrat is the automatic method for error type classification. At least a small qualitative study should have been performed to analyse, whether the errors are well classified. The authors report the results on the automatically obtained annotations, but the performance measured w.r.t. ground truth is not discussed.




**Reproducibility:**

3: Could reproduce the results with some difficulty. The settings of parameters are underspecified or subjectively determined; the training/evaluation data are not widely available.

**Reviewer Confidence:**

4: Quite sure. I tried to check the important points carefully. It's unlikely, though conceivable, that I missed something that should affect my ratings.

---

> ### Author Rebuttal · Authors · 2023-08-26
>
> Thank you very much for your comments and for pointing out the strengths of our work. We address all the points you mentioned in the reasons to reject and the questions you have below.
>
> **Reasons to Reject**:
>
>   * **The effect of including the morphological analysis and error classification is rather small (measured on the test sets, Table 6). Most of the results are reported as not statistically significant**: While the improvements on the test sets are relatively small when using morphological preprocessing, GED, or both, this does not necessarily indicate that using morphology or GED does not help Arabic GEC significantly. In fact, we show on the development sets that when using an Oracle GED system with morphological preprocessing (Table 5), we are able to get an improvement of up to ~5 points on average F0.5.
>
>
> * **Only the overall (average) improvements of the models combining GED and morphological analysis is provided**: We provide results of all systems we develop across all datasets in terms of precision, recall, and F0.5. We only use the average F0.5 in Tables 4 and 5 when we report results on the development sets of QALB-2014, QALB-2015, and ZAEBUC. Our intention of using average F0.5 was merely to select a single best system (AraBART+Morph+GED) when we report results on the test sets individually. That’s why Table 6 does not include average F0.5 results. We will add the averages for the F0.5 test results in the camera-ready for completeness.
>
> * **It is not clear, whether there are some error types where the models can benefit from knowing the explicit information of the type of the error:** In Section 7 (**Lines 534-537**), we mention that we provide in **Appendix E** a specific error type performance by comparing AraBART to our best system that uses morphological preprocessing and GED tags as auxiliary input (AraBART+Morph+GED) on the dev sets. Table 11 in **Appendix E** indicates the specific error types that benefit from modeling GED explicitly.
>
> * **It is not clear, from the paper, how accurate is the automatic method for error type classification. At least a small qualitative study should have been performed to analyse whether the errors are well classified. The authors report the results on the automatically obtained annotations, but the performance measured w.r.t. ground truth is not discussed:** The automatic error type classification tool we use, ARETA ([Belkebir and Habash, 2021](https://aclanthology.org/2021.conll-1.47.pdf)), was evaluated against the manual annotation of the Arabic Learner Corpus (ALC). [Belkebir and Habash, 2021](https://aclanthology.org/2021.conll-1.47.pdf) report that ARETA matches the ALC annotation with 89.2% (F1 micro) on the development test, and 85.8% (F1 micro) on a blind test set. We will make sure to reference the performance of ARETA in the camera-ready of our paper if it’s accepted.
>
>
> **Questions:**
>
>  * **The problem of Arabic dialects is not really discussed in the paper. The large pretrained models are probably trained also on dialectal data. Are dialectal words considered as errors in MSA texts?** Regarding the pretrained Seq2Seq models (i.e., AraBART and AraT5), we mention in **Lines 386-390** that AraBART was pretrained on MSA text while AraT5 was pretrained on a mix of MSA text and Twitter data, which contains dialectal text. As mentioned in Section 3.1, Arabic is diglossic, and borrowing from the dialects is frequent when native speakers write in MSA. The two L1 datasets we use to train and evaluate our models, QALB-2014 and ZAEBUC, contain errors that are almost always in MSA. However, they have errors that are related to dialectal usage. The creators of QALB 2014 intentionally excluded dialectal sentences; and focused on MSA, which may include some dialectal effects. When QALB-2014 was originally annotated, only three types of dialectal errors were manually corrected: morphological choices, phonological choices, and closed-class dialectal words ([Zaghouani et al., 2014](http://www.lrec-conf.org/proceedings/lrec2014/pdf/956_Paper.pdf)). However, these errors were not marked as dialectal errors (in fact no errors are marked at all in QALB data beyond correcting them). The error annotation tool we use was built based on an MSA-only error taxonomy (**Line 288**), and therefore, we do not mark dialectal words using error types outside of the seven error types that are provided by ARETA (**Lines 293-298**). We are aware of other existing Arabic dialectal GEC datasets ([Khalifa et al., 2018](https://aclanthology.org/L18-1607.pdf), [Eryani et al., 2020](https://aclanthology.org/2020.lrec-1.508.pdf)), but our work is mainly intended to address the problem of GEC in MSA. Arabic GEC has a long way to go, and we definitely plan to explore GEC in other Arabic dialectal varieties in future work. We will make sure to mention this in the future work section of the camera-ready version if our paper is accepted.
>
>
>  * **Please, provide better description of the alignment algorithm. It is not clear, whether the Levenshtein algorithm is applied to tokens or to the whole sentences. The greedy part of the algorithm is not explained well**: Our alignment algorithm is similar to a standard word edit-distance alignment algorithm applied at the sentence level (over tokens) with backtracking, except that we have two design enhancements to arrive at multi-token alignments and use character-level information to make optimized word alignment decisions. Our algorithm has 3 main components:
>
>       1.  We modify the standard Levenshtein algorithm such that the token replacement costs are obtained with a secondary application of the Levenshtein edit distance at the token level (over characters).
>       2.  Once we obtain the edit distance table from step 1, we obtain an initial token level alignment by backtracking. Each edit in this alignment (replacement, insertion, deletion, keep) involves one token at most. To obtain multi-token edits, we implement an iterative greedy algorithm, which we explain in 3.
>       3. We apply an iterative greedy algorithm to convert adjacent sequences of replacements, insertions, and deletions into singular operations of word merge or word split such that the overall cost is minimized (based on Levenshtein character-level edit distance).
>
>       We will make sure to expand the description of our algorithm in the camera-ready version if our paper is accepted.
>
>   * **Line 565 refers to a statistical test of significance. Please, provide details on the test results also for other experiments (mainly presented in Table 6):** We did not run standard statistical significance checks for the test sets results of QALB-2015-L2 and ZAEBUC because they are too small to allow us to test for statistical significance (**Lines 621-623**). We will consider Oracle segmentation (based on the corrected gold target) of QALB-2015-L2 and ZAEBUC. This allows us to obtain smaller segments and increase the number of examples in each dataset so we can run statistical significance.
>
>
>
> Thank you again for your comments and suggestions. We hope that our response clarifies your concerns, and we will make sure to address all of them in the extra page we are allowed to add in the camera-ready version if our paper is accepted.

---

### Meta-Review · Area_Chair_vpeC · 2023-09-18

**Recommendation:** 5

**Metareview:**

This paper tackles an underexplored problem in the NLP community, namely the grammatical error correction (GEC) for Arabic language. The paper proposes two Transformer-based models and demonstrated state-of-the-art performance on multiple shared task datasets. The reviewers all agreed on the soundness and excitement of this work.

---

### Decision · Program_Chairs · 2023-10-07

**Decision:**

Accept-Main

**Comment:**

This paper tackles an underexplored problem in the NLP community, namely the grammatical error correction (GEC) for Arabic language. The paper proposes two Transformer-based models and demonstrated state-of-the-art performance on multiple shared task datasets. The reviewers all agreed on the soundness and excitement of this work.